# The diversity of cellular systems involved in carbonate precipitation by *Escherichia coli*

Matthew E. Jennings[1][ID]*, George J. Breley[2], Reilly S. Blackwell[3], Anna Drabik[3], Kathleen R. Gisser[4], Joe Kainrad[2], Victoria Ligman[5], Jaycie Proctor[5], Rose Deshler[ID][5], Brian A. O'Hart[5], Arlyn Rivera[1], Lorelei Centrella[1], Sara Bonn[1], Bisma Chaudhry[1], Hazel A. Barton[ID][3]

1 Department of Biology and Earth System Sciences, Wilkes University, Wilkes-Barre, Pennsylvania, United States of America, 2 Department of Biology, University of Akron, Akron, Ohio, United States of America, 3 Department of Geological Sciences, University of Alabama, Tuscaloosa, Alabama, United States of America, 4 The Sherwin-Williams Breen Technology Center, Cleveland, Ohio, United States of America, 5 Biology Department, Centenary College of Louisiana, Shreveport, Louisiana, United States of America

☯ These authors contributed equally to this work.
* matthew.jennings@wilkes.edu

## Abstract

Climate change is increasing the need to limit levels of anthropogenic $CO_2$ released into the atmosphere. One approach being investigated is to generate products based on microbially induced carbonate precipitation (MICP), which can trap $CO_2$ as $CaCO_3$. We recently identified a novel MICP pathway in bacteria that is initiated by $Ca^{2+}$ toxicity in cells, causing extracellular $CO_2$ to be trapped as $CO_3^{2-}$ by *Escherichia coli*, although the yield of precipitated $CaCO_3$ remained low (in the milligram range). In this work, we used the *E. coli* Keio gene knock-out library to identify 54 genes involved in MICP in *E. coli*, which could be broadly characterized into four groups: central metabolism, iron metabolism, cell architecture, and transport. The role of central metabolism appears to be crucial in maintaining alkaline conditions surrounding the cell that promote $CaCO_3$ precipitation. The role of iron metabolism was less clear, although the results suggest that growth rate influences the initiation of MICP. While the impact of repeating polymeric structures on cell surfaces promoting MICP is well established, our results suggest that other structural features may play a role, including fimbriae and flagella. Finally, the results confirmed that $Ca^{2+}$ transport is central to MICP under calcium stress. The results further suggest that the ZntB efflux pump may play a previously unidentified role in $Ca^{2+}$ transport in *E. coli*. By overexpressing some of these genes, our work suggests that there are several previously unidentified cellular mechanisms that could serve as a target for enhanced MICP in *E. coli*. By incorporating these processes into MICP pathways in *E. coli*, it may be possible to increase the volume of $CO_2$ fixed using this pathway and yield potentially new products that can replace $CO_2$ intensive products, such as precipitated calcium carbonates (PCCs) for industry.

**Data availability statement:** Raw data submitted to Dryad Digital Repository: http://data-dryad.org/share/gYBwYGHo5RKerZ046W1OY-2w60n-pD_dFwNYx7GXoCfw.

**Funding:** HB, KG Grant# HR0011-18-9-0007 DARPA ELM: Engineered Living Materials https://www.darpa.mil/research/programs/engi-neered-living-materials The funders had no role in the study design, data collection and analysis, decision to publish, or preparation of the manuscript. HB, KG, MJ Grant# IIP-2122799 National Science Foundation PFI Grant https://new.nsf.gov/funding/initiatives/pfi The funders had no role in the study design, data collection and analysis, decision to publish, or preparation of the manuscript.

**Competing interests:** The authors have declared that no competing interests exist.

## Introduction

Anthropogenic emissions of greenhouse gases leading to increasing global temperatures represent one of the greatest threats to our biome, with impacts including an increase in extreme weather events, reduced biodiversity, and damage to important ecosystem services [1–5]. The push to find solutions to remove excess anthropogenic $CO_2$ include a range of geological and biological sequestration approaches, such as deep burial and forestry management [6,7]. There is also an increased interest in utilizing carbonatogenesis by microorganisms, which allows the capture of carbon dioxide as $CO_3^{2-}$, which is subsequently precipitated as $CaCO_3$ to prevent release back into the atmosphere [8,9]. Such microbially driven carbonate precipitation has been adapted to various biotechnology applications, including cement-based materials, algal biofuels, and soil amendments [10–14].

Biologically induced mineralization (BIM) occurs when an organism indirectly causes the precipitation of minerals by altering the chemistry of the surrounding environment through metabolic activity, or acts as physical nucleation points for crystal formation [10]. Carbonatogenesis resulting from these processes is known as microbially induced $CaCO_3$ precipitation (MICP), which can occur through a variety of pathways, including ureolysis, photosynthesis, sulfate reduction, nitrate reduction, ammonification, and methane oxidation [15]. Metabolic activity in MICP often alters the local pH, which shifts the dynamic equilibrium of bicarbonate in solution to favor $CO_3^{2-}$ production. This causes them to exceed their saturation index (SI) and precipitate with a divalent cation (usually environmentally abundant $Ca^{2+}$) [16]. Most MICP used in industrial applications use microbial ureolysis, where urease cleaves urea into $CO_2$ and $2NH_4^+$; as a weak base, $NH_4^+$ increases the extracellular pH, driving the precipitation of $CaCO_3$ [17]. This urease-dependent method has been favored in industry due to its ease of use and cost effectiveness, although limitations include the availability of urea and an unpleasant odor [18,19].

Particulate (μm scale) calcium carbonate (PCC) has a number of industrial uses: in paper, carbonates are used as a filler and coating, which can increase the smoothness, brightness and help preserve paper; in thermoplastics, as a filler to reduce polymer volume and modulate elasticity; in sealants and adhesives as a thixotropic agent to reduce shrinkage as polymers set; and in coatings, where carbonates play an important role in opacity, brightness, gloss and durability [20–22]. Over 130,000 kilotons of PCCs were produced worldwide in 2019 in two primary forms [23]: 1) ground calcium carbonates from the mining of calcitic rocks (limestone, marble, travertine, and chalk); and 2) chemically precipitated PCC from CaO (produced by calcination at >1,000°C) reacted with $CO_2$. Both approaches require the mining of calcitic rock, with an increasing pressure on ecosystems and the groundwater found within these landscapes, while the production of PCCs contributes to significant global $CO_2$ emissions [8,24].

Recently, we were able to demonstrate PCC production in liquid cultures of *Escherichia coli* that sources the $CO_3^{2-}$ from atmospheric $CO_2$, via a novel calcium stress-dependent MICP mechanism first described in microorganisms from calcitic cave environments [25]. This pathway relies on the use of potentially toxic levels of

$Ca^{2+}$ via growth on media with a calcium carboxylate salt (calcium acetate, calcium propionate, etc.) [26]. These PCCs have the potential to sequester significant amounts of atmospheric $CO_2$, but also remove the need for energy intensive methods in their production and transport, and would theoretically sequester >1-ton $CO_2$/ton metabolic PCCs (mPPCs) produced [24].

While there are numerous industrial applications for a mPCC product, the current yield of PCC from *E. coli* cultures is low (~100 mg PCC/ 25 mL) containing 62.5 g of $Ca^{2+}$ ions. We therefore decided to investigate whether $CaCO_3$ precipitation in *E. coli* could be enhanced by genetic modification using the Keio collection; this strain library contains non-polar mutations in 4,000 non-essential *E. coli* genes [27]. By screening the Keio library for genes that affected precipitation under calcium stress on calcium propionate, we identified over 50 potential genes involved in a myriad of cellular activities, including central metabolism, cell structure, and transport. The data confirmed that the ability to utilize the organic calcium salt is critical to calcium-stress MICP, while central metabolism plays an important role in carbonatogenesis, from limiting the acidic products of fermentation and cellular $CO_2$ levels. While PCC production in *E. coli* increases the range of industrial applications where products for carbon sequestration could be used, it remains unclear whether any of the genetic modifications identified could produce the high levels of $CaCO_3$ necessary to scale these green PCCs as an industrial replacement to those currently in use.

## Materials and methods

### Bacterial strains and growth conditions

*E. coli* strain BW25113 [28] or K-12 were used as the wild type strain when assaying the various carbonate precipitation phenotypes. The strains were maintained on LB agar plates at 37°C. Single gene deletion strains were obtained from the Keio collection, and consist of single gene mutations in the BW25113 background strain [27]. Single gene deletion strains were maintained on LB agar plates with 50 μg/mL kanamycin at 37°C.

The precipitation of $CaCO_3$ was assayed on modified B4 media. Standard B4 media contains 4 g yeast extract, 10 g glucose, and 15 g agar per liter of media adjusted to pH 7.2 before autoclaving [29]. We used a minimal B4 formulation (B4m), which was prepared as B4 media, but glucose was omitted, with 2.5 g of the relevant calcium salt dissolved in water and filter sterilized before addition to the media following autoclaving. The calcium sources used were calcium acetate (B4m), calcium propionate (B4mPr), calcium L-lactate hydrate (B4mLa), calcium succinate monohydrate (B4mSu), and calcium pyruvate (B4mPy). To improve the solubility of calcium succinate, it was dissolved in 0.1M HCl and readjusted to pH 7.2 before filtering. All solid media was prepared with 1.5% agar. To measure the change in pH in the media upon growth, a single line of WT, ΔnuoA, ΔsdhC, ΔatpF, and ΔprpD were streaked onto separate media plates and incubated at room temperature for 24 hours. The pH was measured using an Orion pH meter with an internal reference probe (Fisher 9863BN) inserted in the agar directly adjacent to the line of growth in three separate locations.

To assay the Keio collection for differences in carbonate precipitation, strains were plated onto B4mPr agar. Media was poured into sterile Nunc OmniTray dishes (Thermo Scientific, Rochester, NY) and allowed to solidify. A 96-pin inoculator was used to spot inoculate 96 strains onto a single plate. Plates were incubated one week at room temperature, and then examined for carbonate precipitation using an Olympus BX53 light microscope (Olympus Life Science, Center Valley, PA) at 100x magnification. Carbonate precipitation was qualitatively compared to WT (BW25113) to identify strains that lacked the carbonate precipitation phenotype.

The iron amendment experiments were carried out in *E. coli* K-12 established in 100 mL B4mSu in a 250mL Erlenmeyer flask. The media was inoculated with 100 μL of an overnight culture of *Escherichia coli* MG1655 (K-12; American Type Culture Collection #700926) and stirred continuously on a Corning PC410-D stir plate with a 3 cm stir bar at 170 rpm. The pH was logged continuously using a REED R3000SD pH meter with a REED R3000SD-PH2 General Purpose pH Electrode. Prior to each experiment, the electrode was calibrated according to the manufacturer's instructions, sterilized by immersion for twenty minutes in 1M HCl, and then washed in sterile DI water.

 

## Cloning and gene expression

Genes were amplified from *E. coli* strain K-12 using standard techniques. Primers were designed using the design primer tool in Geneious Prime 2022.0 (GraphPad Software, Boston, MA), to avoid self-annealing sequences. Forward primers were designed to contain an EcoRI site at the 5' end, and reverse primers contained an XbaI site at the 5' end to facilitate cloning. All primers used in this study were purchased from IDT. A list of primers used for amplification of the native *E. coli* genes can be found in S1 Table. PCR was carried out using Phusion high fidelity polymerase (New England Biolabs, Ipswich, MA) and the PCR products were purified using a Wizard SV PCR Clean-up Kit (Promega, Madison, WI) and quantified using a NanoDrop spectrophotometer (Thermo Fisher, Waltham, MA). Each PCR fragment was cloned into plasmid pPRO24 [30], which contains a propionate driven promoter ($P_{prpB}$). Plasmid and PCR products were both digested with EcoRI and XbaI restriction enzymes (New England Biolabs) prior to ligation, and purified as described above. Ligation products were used to transform *E. coli* strain DH5α (Thermo Fisher), and successful transformants were sequenced to confirm the presence of each gene without mutations. Sequencing primers are listed in S1 Table. Plasmids were then used to transform *E. coli* DE3 cells (Sigma Aldrich, St. Louis, MO) to generate expression strains. Strains containing the plasmid were maintained in the presence of 100 µg/mL ampicillin. Induction of gene expression on plates was achieved by adding 2.5 g/L Na-propionate to the media. Induction of gene expression in liquid media was achieved by adding 50 mM Na-propionate, as not to confound the $CaCO_3$ precipitation experiments.

## Quantification of carbonate precipitation in liquid media

Quantification of solid $CaCO_3$ precipitates in liquid media, 5 mL cultures of carbonate precipitation media were inoculated with 60 µL of an overnight culture of *E. coli* strains grown in LB. The cultures were incubated 24 hours at 37°C. The cultures were harvested by spinning at 5,183 x *g* in an Avanti J-E centrifuge for 10 minutes. The supernatant was removed, and pellets washed 1X in 1 mL 1x PBS pH 8. The pellet was resuspended in 1 mL of 5% $HNO_3$ to dissolve any precipitated calcium carbonate for 30 minutes at room temperature [25]. The supernatants were assayed using the Arsenazo III calcium sensitive dye (Pointe) at 650 nm on a Molecular Devices Spectramax M4 spectrophotometer. A standard curve was generated using soluble calcium (Inorganic Ventures 10PPM) at 4, 6, 8, 10, 12, and 15 mg/dL. There was no significant difference in $CaCO_3$ production between *E. coli* BW25115 and DE3 in this assay.

## Preparation of SDS-PAGE gels

Expression studies were done using twenty-five mL of LB media inoculated with 250 µL of an overnight culture of the relevant strain. The culture also contained 100 µg/mL ampicillin to maintain the plasmid. Flasks were incubated at 37°C and growth was monitored via absorbance at 600 nm, and cultures were induced with 50 mM Na-propionate when absorbance reached 0.6. A separate culture for each strain was not induced. Flasks were then incubated at 20°C for 12 hours. Cultures were harvested by centrifugation for 10 minutes at 5,180 xg in a Beckman-Coulter Avanti J-E centrifuge. Cell pellets were resuspended in 1 mL TEN buffer (10 mM Tris-HCl, 10 mM EDTA, 150 mM NaCl, pH 8.0) and a small amount of benzamidine and phenylmethylsulfonyl fluoride (PMSF) were added to inhibit proteases. Cells were lysed using a Qsonica Q55 Sonicator set to 100 Hz. Each sample was sonicated five times, with each sonication lasting 30 seconds. Following sonication, cell samples were spun at 11,180 xg for 10 minutes in a Thermo-Scientific Sorvall ST 16R centrifuge at 4°C. The soluble fractions were separated from insoluble fraction using a micro pipettor, then quantified using a Qubit 2.0 fluorometer system. The insoluble fractions were resuspended in 2 mL 2x SDS gel loading buffer (100 mM Tris-HCl, 4% SDS, 0.2% bromophenol blue, 20% glycerol, 200 mM β-mercaptoethanol). Insoluble fractions were sonicated twice as described above, and then boiled for 10 minutes prior to loading. Samples were loaded onto a Thermo-Fisher Invitrogen Novex 12% Tris-Glycine protein gel (XP00120BOX). Soluble fractions were normalized based on total protein and insoluble fractions were loaded based on total volumes. Gels were run in an Owl P81 electrophoresis chamber with the voltage

set to 100 V. Gels were stained with Coomassie Blue (450 mL methanol, 450 mL water, 100 mL glacial acetic acid, 2.5/g Coomassie Brilliant Blue R-250) for 1 hour. Gels were destained overnight (500 mL methanol, 400 mL water, 100 mL acetic acid) and then imaged the next day.

## Results

### Identification of the genes involved in carbonate precipitation

To identify genes associated with $CaCO_3$ precipitation, we spot-plated knockout strains from the Keio *E. coli* library [27] onto a calcium propionate medium lacking glucose (B4mPr) and monitored for 1 week at room temperature. We previously demonstrated that this media is an effective screen for carbonatogenesis in *E. coli* [25], with the formation of visible $CaCO_3$ crystal after 4 days at room temperature [25]. To demonstrate the effectiveness of this approach, the screen identified *prp*D, which functions in the 2-methylcitrate pathway of propionate catabolism, and cells lacking this gene cannot utilize calcium propionate for growth and thus produce no $CaCO_3$ crystals (Fig 1) [31]. Of the ~4,000 strains screened, we observed 54 mutants with a WT growth phenotype, but lacking a $CaCO_3$ precipitation phenotype. This included a variety of known cellular functions that could be clustered into 4 major groups: central metabolism, iron metabolism, cell architecture, and transport (Table 1). Eleven additional genes with miscellaneous cellular functions were also identified, including the *prp*D control. One cytoplasmic membrane protein of unknown function (*ygd*Q) was identified, which was not investigated further (Table 1). All knockout strains that were identified were re-streaked to confirm the loss of the carbonatogenesis phenotype.

### The role of central metabolism

Of the genes identified in central metabolism, all were associated with protein complexes in the tricarboxylic acid (TCA) cycle and electron transport (Table 1 and Fig 1): succinate:quinone oxidoreductase, SQR (*sdh*CDAB); α-ketoglutarate dehydrogenase (*suc*ABCD); NADH:quinone oxidoreductase, Complex I [*nuo*ABC(D)EFGHIJKLMN]; and ATP synthase (*atp*ABCDFH). The fact that every representative gene from each of these operons was identified in our screen of the Keio library confirms the role that these pathways play in carbonatogenesis on propionate (Table 1). The lack of major components of the TCA cycle and electron transport can affect metabolism in a variety of ways, so we decided to examine whether the observed effects were related to growth rate or specific to carbonatogenesis. In the case of α-ketoglutarate dehydrogenase, mutations in the *suc* operon have been shown to reduce the amount of $CO_2$ released by the TCA cycle by >50% [32]. Given the need to create $CO_3^{2-}$ ions for $CaCO_3$ precipitation, we felt that it was not necessary to explore the impact of the *suc* knockouts further. We randomly chose a representative gene from Complex I, SQR, and ATP synthase (Δ*nuo*B, Δ*sdh*C, and Δ*atp*F) to understand the loss-of-function impact of these systems on carbonatogenesis, along with WT and Δ*prp*D as controls (Table 1).

The use of solid media allows us to rapidly assess $CaCO_3$ precipitation by visual inspection, but it also allows us to examine some of the local conditions that are difficult to measure in liquid culture, such as the distribution of pH gradients within the environment [25]. We have previously shown that growth of WT *E. coli* in B4-m media produces basic conditions, regardless of the calcium source, presumably through the consumption of amino acids and export of $NH_3$ into the surrounding media [25]. When the pH approaches 8.3, the ion activity product exceeds the SI for $Ca^{2+}$ and $CO_3^{2-}$, leading to $CaCO_3$ precipitation [25]. As has been previously shown, on all of the calcium sources used, WT *E. coli* raised the pH (range 8.1–8.5) and led to the precipitation of $CaCO_3$ (Fig 2A) [25]. The cells need to catabolize the organic calcium salt for carbonatogenesis (Fig 1) [26], and unsurprisingly, the Δ*prp*D knockout only prevented $CaCO_3$ precipitation on calcium propionate (Fig 2A).

the initiation of carbonatogenesis (indicated in WT by the black arrow). The dashed line indicates the pH leads to calcium carbonate exceeding the saturation index and begins precipitating.

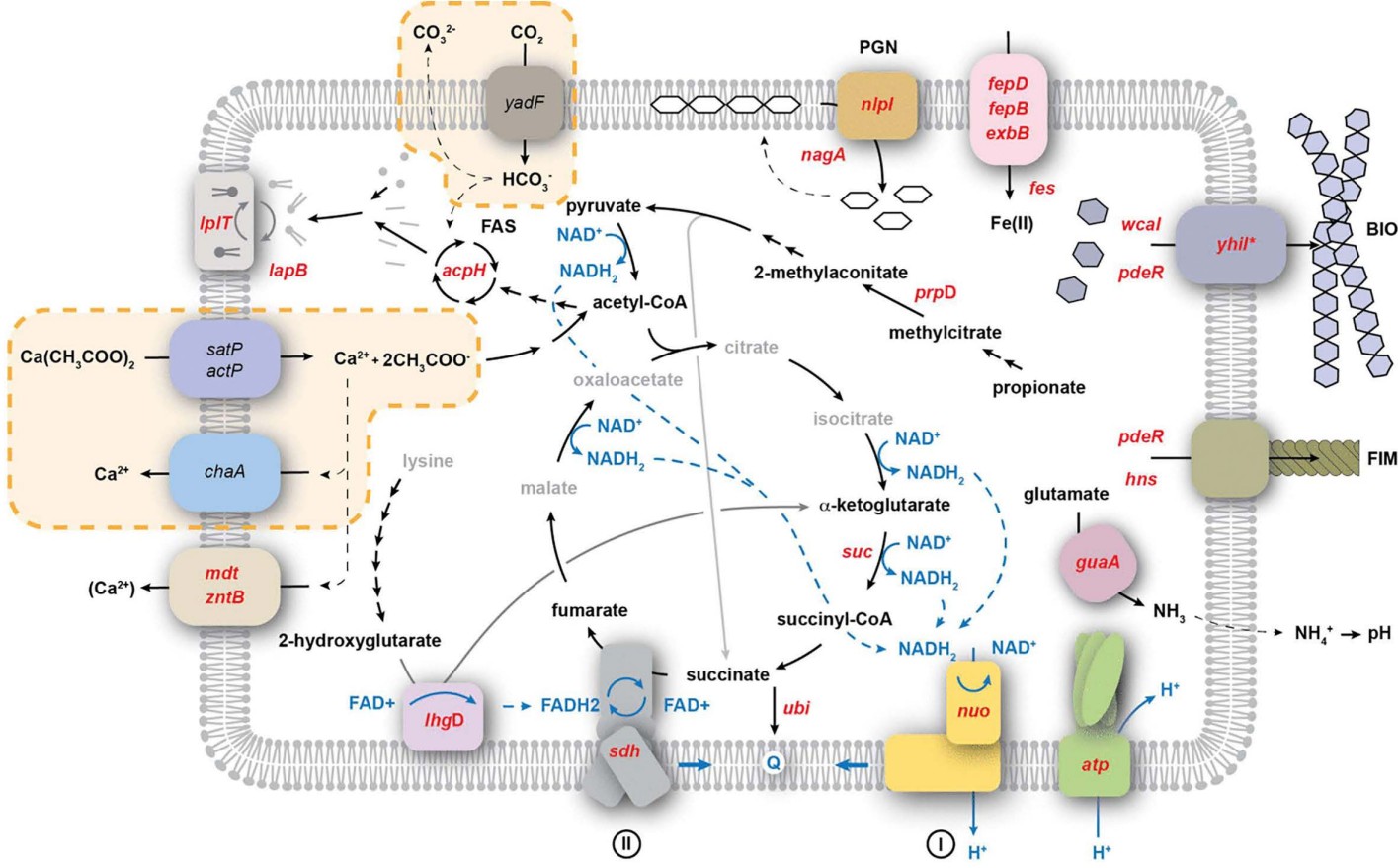

**Fig 1. Metabolic pathways influencing carbonatogenesis identified in this study.** The calcium homeostasis pathway that drives carbonatogenesis in our model is highlighted in the yellow dashed boxes: uptake of the organic calcium salt (represented by calcium acetate) via the sat or act operons leads to the metabolism of the carboxylic acid (the glyoxylic acid pathway for acetate metabolism within the TCA is not shown for simplicity), with the excess $Ca^{2+}$ removed via an unknown calcium transporter. The uptake of $CO_2$ via the carbonic anhydrase (*yadF*) provides the source of $CO_3^{2-}$ ions for precipitation. The Keio library knockout strains identified with defects in carbonatogenesis are shown in red. While ChaA was previously demonstrated to be the $Ca^{2+}$ transporter in *Salmonella* [26], our screen indicated a phenotype for the transporters *mdtH*, *mdtL*, and *zntB*. The intermediates shown in central metabolism and metabolism of the provided calcium sources are shown in black (central metabolites not identified with a role in carbonatogenesis in this work are gray). Blue indicates the flow of electrons and reducing equivalents. The cell wall and membrane structure of *E. coli* is represented as a single membrane for simplicity.

The loss of *sdhC* prevents the assembly of the SQR complex, which catalyzes the oxidation of succinate to fumarate coupled to the reduction of FAD in the TCA cycle [33]. The lack of this enzyme prevents proper functioning of the TCA cycle; however, the enzyme fumarate dehydrogenase, which catalyzes the reduction of fumarate to succinate, can operate in the reverse direction, albeit at reduced efficiency with secretion of acetic acid [34,35]. Our data indicated that the Δ*sdhC* knockout acidifies the surrounding medium under all conditions tested (Fig 2A), presumably through the production of this acid. Growth in liquid cultures allows us to qualitatively evaluate the growth kinetics of each strain, and when carbonatogenesis occurs it can be observed by a notable increase in the $OD_{600}$ signal (via light scattering by the formation of $CaCO_3$ particulates) and a simultaneous drop in pH (as $CO_3^{2-}$ is consumed) [25]. These data (Fig 2B) indicate that the overall growth rate of Δ*sdhC* was slower that WT (mid-log doubling time 1.7 hrs, compared with 1.27 hrs for WT), with the pH only minimally exceeding the SI for $CaCO_3$ after 30 hrs of growth (Fig 2C).

**Table 1. List of *E. coli* Keio knockout strains that lack the CaCO$_3$ precipitation phenotype; grey highlighted genes were chosen for further characterization.**

| Strain ID | Gene | Gene function | Blattner # |
|---|---|---|---|
| **Central Metabolism** | | | |
| JW0713 | *sdhA* | succinate:quinone oxidoreductase | b0723 |
| JW0714 | *sdhB* | | b0724 |
| JW0711 | ***sdhC*** | | b0721 |
| JW0712 | *sdhD* | | b0722 |
| JW0715 | *sucA* | α-ketoglutarate dehydrogenase | b0726 |
| JW0716 | *sucB* | | b0727 |
| JW0717 | *sucC* | | b0728 |
| JW0718 | *sucD* | | b0729 |
| JW2283 | *nuoA* | NADH:quinone oxidoreductase | b2288 |
| JW5875 | ***nuoB*** | | b2287 |
| JW5375 | *nuoC* | | b2286 |
| JW2280 | *nuoE* | | b2285 |
| JW2279 | *nuoF* | | b2284 |
| JW2278 | *nuoG* | | b2283 |
| JW2277 | *nuoH* | | b2282 |
| JW2276 | *nuoI* | | b2281 |
| JW2275 | *nuoJ* | | b2280 |
| JW2274 | *nuoK* | | b2279 |
| JW2273 | *nuoL* | | b2278 |
| JW2272 | *nuoM* | | b2277 |
| JW2271 | *nuoN* | | b2276 |
| JW3712 | *atpA* | ATP synthase F1 complex | b3734 |
| JW3716 | *atpB* | | b3738 |
| JW3709 | *atpC* | | b3731 |
| JW3710 | *atpD* | | b3732 |
| JW3714 | ***atpF*** | | b3736 |
| JW3713 | *atpH* | | b3735 |
| **Iron Metabolism** | | | |
| JW0576 | *fes* | Esterase that catalyzes the hydrolysis of both enterobactin and ferric enterobactin to release Fe | b0585 |
| JW0582 | *fepD* | Component of ferric enterobactin ABC transporter | b0590 |
| JW0584 | *fepB* | Component of ferric enterobactin ABC transporter | b0592 |
| JW2974 | *exbB* | Subunit of the Ton complex involved in active transport of iron-siderophore complexes and other cofactors across the outer membrane | b3006 |
| **Cell Architecture** | | | |
| JW0663 | *nagA* | N-acetylglucosamine-6-phosphate deacetylase that catalyzes the first cytoplasmic reaction in the metabolism of N-acetyl-D-glucosamine. | b0677 |
| JW1272 | *lapB* | Protein involved in coordinating LPS and phospholipid biosynthesis | b1280 |
| JW1278 | *pdeR* | cyclic di-GMP phosphodiesterase involved in the production of biofilm extracellular matrix | b1285 |
| JW2035 | *wcaI* | Plays a role in the synthesis of extracellular polysaccharides. | b2050 |
| JW2284 | *lrhA* | Regulator involved in fimbriae and flagella synthesis | b2289 |
| JW2803 | *lplT* | Transporter for movement of lysophospholipids across the cytoplasmic membrane | b2835 |
| JW3132 | *nlpI* | Appears to play a role in coordinating hydrolases within peptidoglycan biosynthetic complexes | b3163 |
| JW3454 | *yhiI* | Putative ABC transporter. Mutations in yhiI leads to abnormal biofilm structure. | b3487 |

*(Continued)*

**Table 1.** (Continued)

| Strain ID | Gene | Gene function | Blattner # |
|---|---|---|---|
| **Efflux Pumps** | | | |
| JW1052 | mdtH | Multidrug efflux pump | b1065 |
| JW1336 | **zntB** | Believed to be a Zn2+/H+ symporter for Zn2+ uptake | b1342 |
| JW1591 | mdtI | Spermidine efflux pump. Spermidine regulates cellular Ca2+ influx. | b1599 |
| **Misc Cell Function** | | | |
| JW0325 | **prpD** | 2-methylcitrate dehydratase involved in propionate catabolism | b0334 |
| JW0382 | ppnP | Nucleoside phosphorylase involved in the stringent response | b0391 |
| JW0394 | acpH | Phosphodiesterase with a role in fatty acid synthesis, although exact role remains unknown | b0404 |
| JW0703 | pxpA | Part of a two component (CPX regulon) that responds to changes in the cytoplasmic membrane. Genes involved include efflux pumps, DNA repair, adherance, and motility. | b0713 |
| JW1225 | hns | Modulates chromatic structure, giving it a broad role in cellular stress responses. Demonstrated roles in flagella synthesis, acid resistance, and growth on a variety of carbon sources | b1237 |
| JW2226 | ubiG | Involved in ubiquinone and menaquinone biosynthesis, along with ubiE | b2232 |
| JW2461 | purC | Component of the purine biosynthesis pathway | b2476 |
| JW2491 | guaA | GMP synthetase that generates ammonia from glutamate | b2507 |
| JW2635 | lhgD(O) | Dehydrogenase that converts L-2-hydroxyglutarate generated from lysine metabolism to a-ketoglutarate with the generation of FADH2 for the ETC | b2660 |
| JW2858 | lysS | Constitutively expressed lysine--tRNA ligase. Mutants have been associated with slower growth. | b2890 |
| JW5581 | ubiE | Involved in ubiquinone and menaquinone biosynthesis. Mutants lack ubiquinone and can primarily use fermentative growth. | b3833 |
| **Unclassified** | | | |
| JW2800 | ygdQ | Cytoplasmic membrane protein of unknown function | b2832 |

The AtpF protein is involved in the formation of the ATP synthase complex, requiring growth on fermentable carbon sources [36,37], and the ΔatpF strain did not produce carbonate regardless of calcium source (Fig 2A). Lactate and pyruvate can be fermented, which would presumably lead to the production of acetic and other acids [38–40]; however, there is no clear difference in the pH of the surrounding media between the use of fermentable (lactate and pyruvate) and non-fermentable calcium sources (acetate, propionate, and succinate) in the media (Fig 2A). The presence of a lag phase and much slower growth for ΔatpF likely corresponds to the shift in cellular metabolism from respiratory to fermentative growth (Fig 2B), and given that the ΔatpF strain produced colonies on non-fermentable calcium acetate, propionate, and succinate, it suggests that the observed growth is dependent on the amino acids present in the media (as for the ΔprpD knockout on propionate; Fig 2A). Nonetheless, while growth in the ΔprpD led to an increased pH in the surrounding media, ΔatpF did not, suggesting a more complex response to the formation of basic conditions during amino acid catabolism (Fig 2A and 2C).

### The role of central metabolism – Complex I

The strain ΔnuoB had reduced levels of $CaCO_3$ precipitation in our initial screen (Table 1). While this mutant raised the pH using all calcium sources tested (pH range 8.0–8.7), it maintained a reduced precipitation phenotype on all media (Fig 2A). Given the critical role of Complex I in electron transport, it was possible that this knockout was simply affecting the growth rate, which in turn would reduce catabolism of the calcium salt and observed levels of $CaCO_3$ precipitation [25,26], but while ΔnuoB grew more slowly in liquid culture, it was more comparable to WT than other assayed mutants (mid-log doubling time 1.51 hrs; Fig 2B).

We decided to explore the slow onset of carbonatogenesis in ΔnuoB in a more quantitative way and assayed the amount of $CaCO_3$ precipitated between ΔnuoB, WT, and ΔprpD in liquid cultures containing the various calcium sources

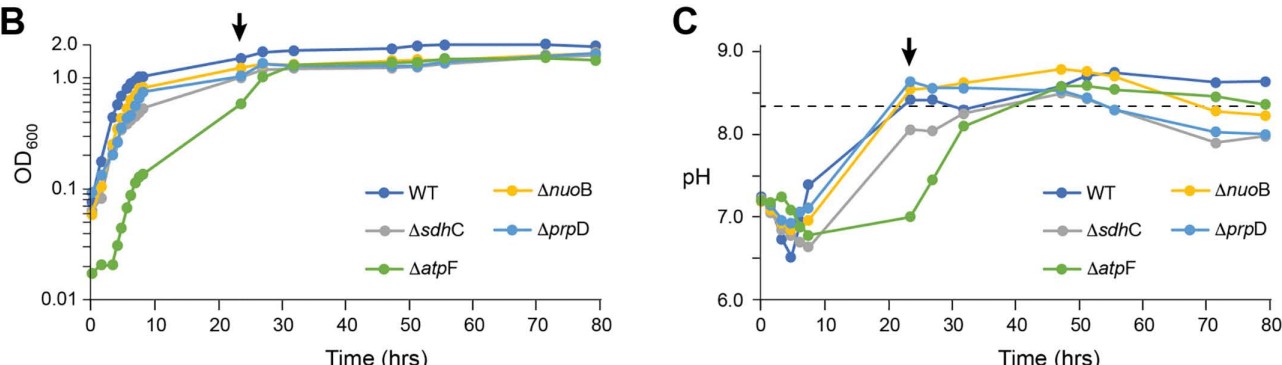

| | | Organic calcium salt | | | | |
|---|---|---|---|---|---|---|
| | | Acetate | Propionate | Lactate | Pyruvate | Succinate |
| **Strain** | WT | 8.4 ± 0.10 | 8.1 ± 0.06 | 8.5 ± 0.10 | 8.3 ± 0.02 | 8.5 ± 0.03 |
| | ΔnuoB | 8.2 ± 0.04 | 8.0 ± 0.08 | 8.6 ± 0.07 | 8.3 ± 0.06 | 8.7 ± 0.07 |
| | ΔsdhC | 6.8 ± 0.05 | 6.2 ± 0.01 | 6.6 ± 0.09 | 6.5 ± 0.40 | 6.8 ± 0.38 |
| | ΔatpF | 7.4 ± 0.08 | 7.7 ± 0.10 | 6.4 ± 0.06 | 7.7 ± 0.19 | 6.5 ± 0.07 |
| | ΔprpD | 8.6 ± 0.20 | 8.1 ± 0.10 | 8.3 ± 0.12 | 8.4 ± 0.04 | 8.4 ± 0.03 |

**Fig 2. Qualitative comparison of calcium source on carbonatogenesis in selected *E. coli* knock-out mutants compared to WT. A)** Strains were spot plated onto solid carbonatogenesis media plates containing the indicated organic calcium-salt. The pH of the media after 24 hours of growth at 37°C was measured (the average of three pH measurements +/- SD). The color of the cell is based on a visual qualification of the amount of PCC produced relative to WT; deep green, high precipitation; pale green, limited precipitation, white, no precipitation; pink indicates the knock-out prevents use of the indicated organic calcium salt for growth. **B)** The growth rate in liquid culture containing calcium propionate as measured by $OD_{600}$. The initiation of carbonate precipitation can be observed by a bump in $OD_{600}$ (indicated in WT by the black arrow). **C)** Change in the pH of the media corresponding to the liquid culture in **(B)**. A drop in pH indicates.

(Fig 3). The amount of $CaCO_3$ in the cultures was measured at 24 hr, when the initiation of precipitation is observed in WT (Fig 2C). The results demonstrated limited precipitation in acetate and propionate between ΔnuoB, WT, and ΔprpD at 24 hrs, which is consistent with prior observations of a lower level of carbonatogenesis in *E. coli* using these calcium sources [25]. When grown on calcium lactate, pyruvate, and succinate, ΔprpD demonstrates the same precipitation phenotype as WT, with the highest amount $CaCO_3$ precipitation when these strains were grown on calcium succinate (Fig 3). The data also demonstrated that ΔnuoB precipitated significantly less $CaCO_3$ than either WT or ΔprpD on calcium lactate, pyruvate, and succinate (Fig 3). These data correlate with the quantitation results seen on the solid media, confirming the effectiveness of the spot plating method as a rapid screen to observe genetic impacts on carbonatogenesis in *E. coli* (Fig 3).

### Iron metabolism

Several genes involved in iron acquisition and transport were identified as affecting carbonatogenesis (Table 1). The relationship of iron acquisition and transport genes to $CaCO_3$ precipitation is not clear, though the issue may simply be poor growth due to limiting iron when these genes are missing [41]. To determine if the precipitation phenotype is related to iron transport, we examined the impact of amending WT cultures with iron. We continuously monitored pH as a proxy for the initiation of $CaCO_3$ precipitation (Fig 4), with a sudden drop in pH corresponding with the onset of $CaCO_3$ crystal accumulation in the bottom of the flask [25]. We performed this assay at room temperature, to slow the rate of growth and provide

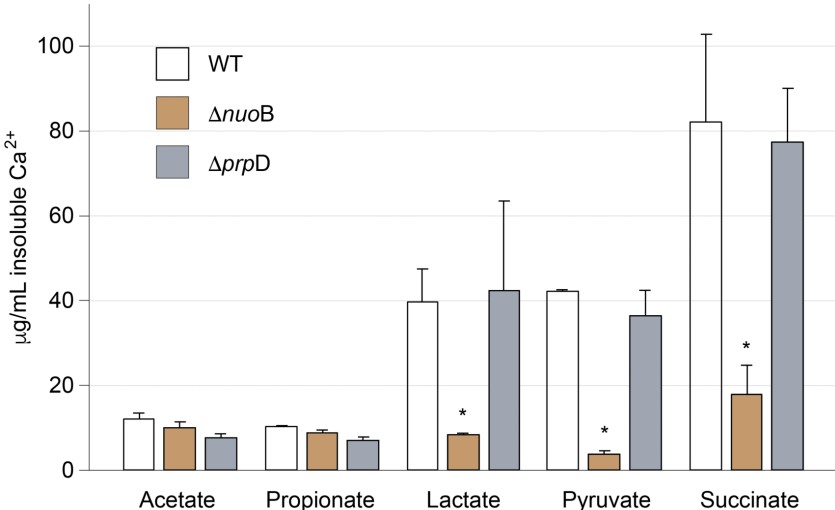

**Fig 3. Quantification of insoluble Ca²⁺ from 25 mL B4m media containing the indicated calcium salt following 24 hr incubation.** Ten mL of each culture was harvested, soluble Ca removed via washing, and then the remaining pellet treated with 5% $HNO_3$ to dissolve $CaCO_3$. Values are average of three replicates and error bars represent standard deviation. An asterisk represents a significant difference compared to the BW25113 values (WT) using a Student's T-test ($p < 0.05$).

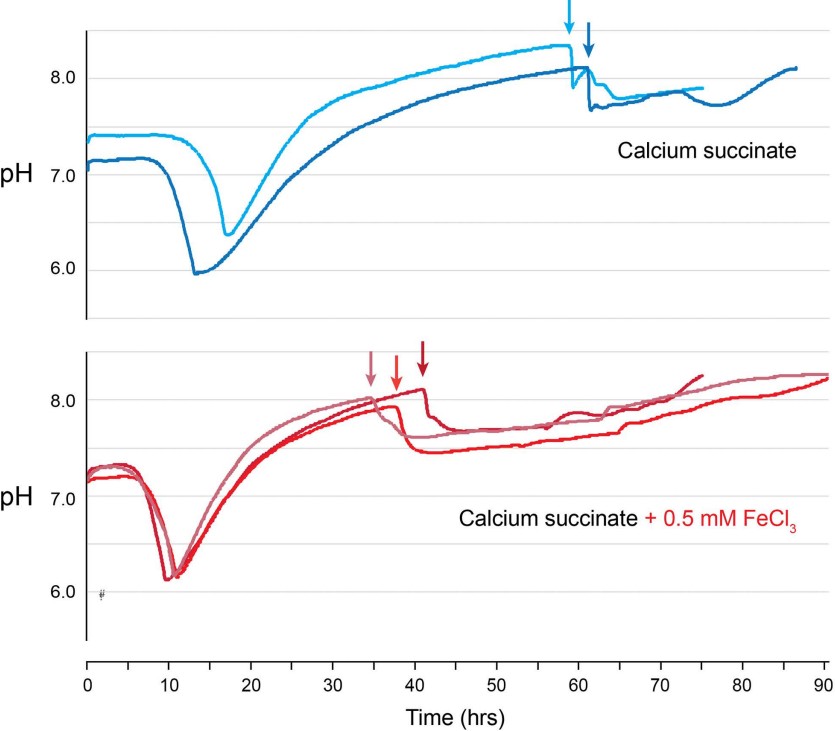

**Fig 4. The impact of iron on $CaCO_3$ precipitation rate.** WT *E. coli* K-12 cultures were grown at room temperature in calcium succinate, with pH used as a proxy for the initiation of $CaCO_3$ precipitation. The pH of the culture was monitored in 1 minute intervals for replicate experiments, with (red data logs) and without (blue data logs) the addition of 0.5 mM $FeCl_3$. Black arrows indicate the initiation of carbonatogenesis for each replicate.

a better resolution of the timing of carbonatogenesis (no significant change in yield between growth at RT and 37°C has been observed previously [25]).

When WT *E. coli* was grown using calcium succinate, precipitation began around 60 hrs (Fig 4), which corresponds with the observations of similar culture conditions [25]. When the media was amended with 0.5 mM $FeCl_3$, the onset of precipitation was reduced to between 32–41 hrs, without a significant change in growth rate (Fig 4), suggesting that Fe(III) amendment does influence the timing of carbonatogenesis. The resulting crystals had a distinctive brown color, suggesting that available Fe(III) was being incorporated into the growing crystals.

## Cell architecture

Our screen similarly suggested that a variety of extracellular structures in *E. coli* influence $CaCO_3$ precipitation (Table 1). Bacteria bind metal ions, such as $Ca^{2+}$ by virtue of the acid/base properties of the cell wall functional groups, which include amino, carboxylic, hydroxyl, and phosphate sites [42,43]. The genes identified in the Keio library screen suggests that there is a myriad of factors that may influence this coordination and nucleation of the forming $CaCO_3$ nanocrystals, including EPS, LPS, and peptidoglycan (Table 1), which have previously been demonstrated to play an important role in nucleating $CaCO_3$ crystals. Nonetheless, our data also identified the potential for the proteinaceous repeating units of fimbriae and flagella in promoting such nucleation, although the experiments needed to study these interactions are beyond the scope of this study.

## Efflux pumps

Our screen identified three other transporters, Δ*mdtH*, Δ*mdtI*, and Δ*zntB* (Table 1), which have not previously been associated with $Ca^{2+}$ transport [44]. The *mdtH* (*yceL*) and *mdtL* (*yidY*) transporters were identified through sequence annotation of the *E. coli* genome as members of the major transporter superfamily involved in antibiotic resistance, although the over expression of *mdtH* or *mdtL* only weakly enhanced resistance [44,45]. No additional work has attempted to understand the cellular function of these transporters [44]. The ZntB transporter protein is believed to be involved in zinc homeostasis, although whether it functions as an importer or exporter remains under debate [46,47].

Given the greater level of understanding of *zntB* function (compared to *mdtH* and *mdtL* [46]), we decided to study its effect on carbonatogenesis by cloning into an expression vector (pPRO24) in *E. coli*, which includes a promoter under the control of the propionate-inducible prpR regulator [30]. We also cloned *yrbG,* which encodes a sodium/calcium ion exchanger that regulates *E. coli* cytosolic calcium, but has no described role in MICP [48]. Along with *zntB,* both genes were inserted into pPRO24 and sequenced to confirm that they had 100% identity to the WT gene.

The expression plasmids (pPRO::*yrbG* and pPRO::*zntB*) were transformed into *E. coli* DE3 and induction of the target proteins, YrbG and ZntB, was confirmed by SDS-polyacrylamide gel electrophoresis (S1 Fig). We then assayed these constructs, along with a plasmid control*,* for $CaCO_3$ production with or without induction with sodium propionate (Fig 5). There was a small, but not significant, increase in $CaCO_3$ precipitation in the plasmid control with the addition of the inducer (25 mM sodium propionate; Fig 5), which did alter protein expression profiles (S1 Fig) and may be due to change in metabolism due to the supplemented sodium propionate (Fig 5). The precipitation profile of the pPRO::*yrbG* demonstrated a small increase in $CaCO_3$ precipitation without induction along with an apparent change in YrbG expression compared to WT, but the increase in $CaCO_3$ was not significant (Fig 5 and S1 Fig). We did see a significant impact on the precipitation phenotype in pPRO::*zntB,* with a 3-fold increase in $CaCO_3$ precipitation in both the uninduced and induced strain (Fig 5). As a transmembrane bound protein, ZntB purifies with the insoluble cell fraction, making expression more challenging to observe via SDS-PAGE. Nonetheless, a slight increase in the expression of a protein at the correct size for ZntB was identified upon induction in the insoluble fraction (S1 Fig). The pPRO24 promoter is known to be leaky, allowing a potential dosing effect from the multicopy plasmid prior to induction [30]. It is also possible that, as a transmembrane bound protein, ZntB has a longer half-life that the membrane-associated YrbG, which may amplify any dosing effects.

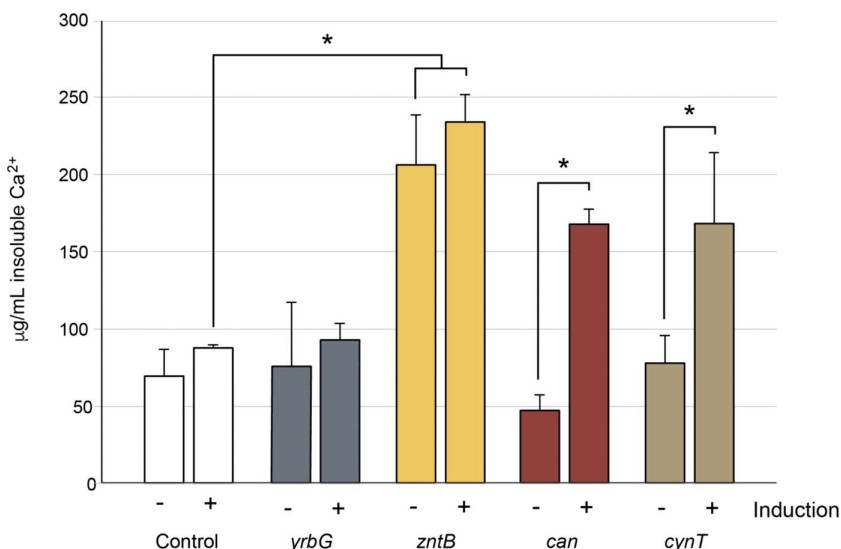

**Fig 5. Quantification of CaCO₃ precipitation from genetic constructs in *E. coli* DE3.** Cells containing the plasmid control (control) or calcium transporter (*yrbG*; which does not play a role in carbonatogenesis) were compared with the transporter *zntB*. Cultures were either uninduced (-) or induced (+) with 25 mM sodium propionate. To confirm that our assay did induce enhanced carbonatogenesis through induction, two genes known to increase CaCO₃ precipitation (can and cynT) we used. Each bar represents the average of three replicate cultures, with bars corresponding to standard deviation. Statistical comparisons were done using a Student's T-test, with a significance of $p < 0.05$ indicated by an asterisk.

Prior research has demonstrated an important role for CA in CaCO₃ precipitation in bacteria, including a periplasmic expressed CA in *E. coli* [49–52]. These genes convert atmospheric CO₂ to bicarbonate and their overexpression has previously been demonstrated to dramatically improve carbonate precipitation [51]. To compare the impact of CA expression with the *zntB*, we cloned the *E. coli* CA genes *can* (*yadF*) and *cynT*. Upon induction, a significant increase in CaCO₃ precipitation was detected, which is consistent with previous studies, and demonstrated the validity of our expression approach [51,53].

## Miscellaneous cell function

A variety of other genes were identified in our knockout assays (Table 1), with a range of cellular functions. While the role of some of the genes in CaCO₃ precipitation may be relatively straightforward (for example, *guaA* releases ammonia from the catabolism of amino acids and raises pH), others may be more challenging to explain. In particular, there were a number of genes involved in the cellular stress response (*ppaP, pxpA, hns*). It is our understanding that the calcium homeostasis phenotype leading to CaCO₃ precipitation is driven to some extent by cellular growth under non-ideal conditions, which may require the cell stress response to maintain active growth [26]. Understanding the role of these genes will require additional investigation beyond the scope of this work.

## Discussion

MICP is an exciting developing area of applied microbiology, both as a potential avenue of carbon sequestration and a way to enhance building materials for self-repair [16]. The most common MICP approach in building materials, such as cements, is ureolysis [17]. But the release of ammonia can lead to odor issues, while ammonia can be oxidized to N₂O, itself is a greenhouse gas [54]. We previously demonstrated carbonate precipitation by *E. coli* using calcium homeostasis (Fig 6) [26], and in this paper we aimed to identify other genes that could enhance this pathway for industrial applications

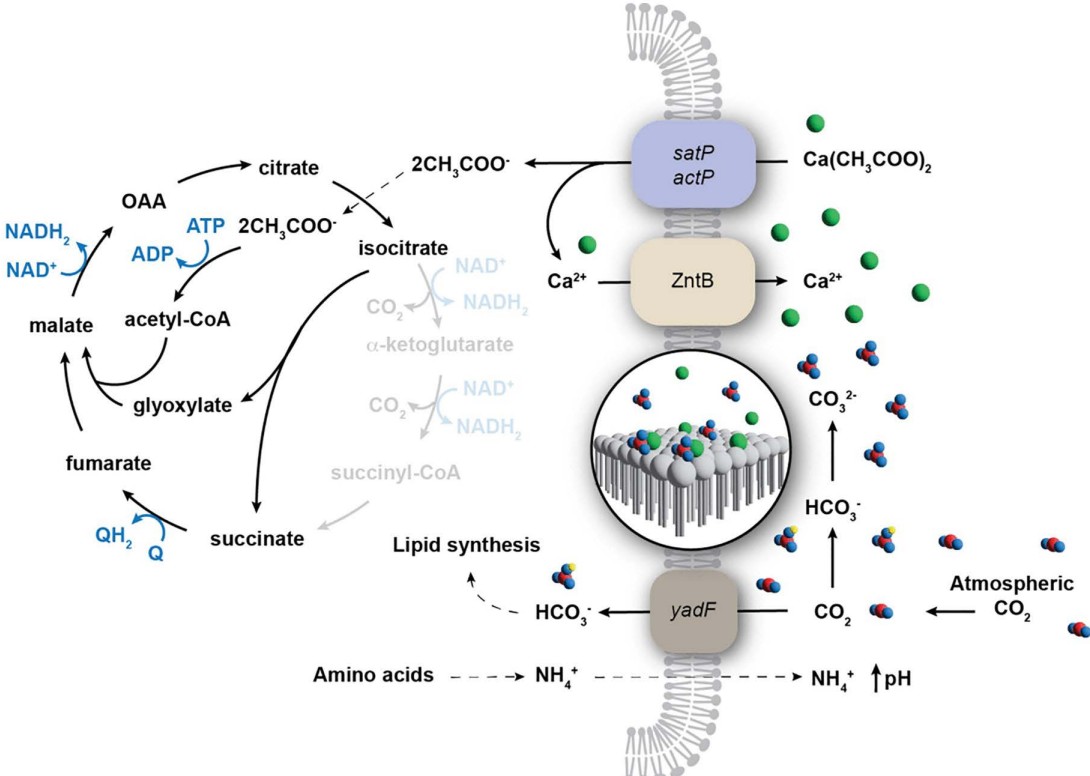

**Fig 6. Revised model for carbonatogenesis via calcium homeostasis.** The initial steps of our revised carbonatogenesis pathway remain unchanged: an organic calcium salt (represented by calcium acetate) is taken up by the cell (via *sat* or *act*), which leads to the metabolism of the carboxylic acid via the glyoxylate shunt of the TCA cycle. Carbonic anhydrase (*yadF/can*) produces bicarbonate from atmospheric $CO_2$ to compensate for the loss of $CO_2$ (the missing portions of the TCA cycle are indicated with light shading) via the glyoxylate shunt. The uptake of $CO_2$ leads to an increase in the surrounding pH. Our data suggest that in *E. coli*, ZntB may be involved in the export of excess $Ca^{2+}$ that are coordinated on the negatively charged surface of the cell, reducing the energy necessary to initiate mineralization. The catabolism of amino acids leads to the release of $NH_4^+$, which additionally raises the pH and increases the relative ratio of $CO_3^{2-}$ ions surrounding the cell. Combined these conditions cause the local environment to favor the precipitation of calcium carbonate, with $CO_2$ taken up in both cellular biomass and the growing carbonate minerals. Blue indicates the flow of electrons and reducing equivalents. The cell wall and membrane of *E. coli* is represented as a single membrane for simplicity.

independent of ureolysis. Our data indicated that there are a wide variety of cellular functions that influence MICP in *E. coli*, from central metabolism to cell structure and the stress response (Table 1). While the specific function of many of these genes still needs to be elucidated, they did provide some new clues to the metabolic pathways that influence $CaCO_3$ precipitation in *E. coli* and allowed us to refine our model of how cellular $Ca^{2+}$ stress drives carbonatogenesis (Fig 6).

Carbonatogenesis via $Ca^{2+}$ homeostasis requires the metabolisms of organic calcium salt (Fig 6) [25,26]. Our mutant screen (Table 1) identified methylcitrate dehydratase (Δ*prpD*), which is involved in the catabolism of propionate, supporting the robustness of our screening approach. Nonetheless, there are four other genes involved in the 2-methylcitrate cycle, which converts propionate into succinate and pyruvate (Δ*prpB*, Δ*prpC*, Δ*prpE* and Δ*acnB*) that were not identified in our screen (Table 1) [55]. The genes Δ*prpB* and Δ*acnB* function downstream of *prpD*, and a knockout in all three genes would lead to an increase in cellular levels of 2-methylcitrate, a co-repressor that downregulates expression of the *prp* operon, further limiting propionate catabolism [56]. The loss of Δ*prpE* and Δ*prpC* did not lead to a loss of carbonatogenesis, but these mutations could allow propionate catabolism to proceed via the propionyl-CoA degradation pathway, allowing the release of the $Ca^{2+}$ ion for carbonatogenesis [57]. An acetyl-CoA ligase (*acs*) can also compensate for the loss of

the propionyl-CoA *prpE* [58]. Although there may be additional complexity in the propionate catabolism pathway that was not captured in this assay, our data support the model that catabolism of the calcium carboxylate is critical for MICP (27).

Our prior work in *Salmonella* also demonstrated that a knockout in the $Ca^{2+}$ transporter ChaA had a dramatic effect on carbonatogenesis; however, we did not identify Δ*chaA* during the Keio library screen [26]. Research has shown that in *E. coli* ChaA has a higher affinity for $K^+$ than $Ca^{2+}$ ions and plays an important role in cell growth under alkaline conditions [59,60]. We did attempt to use *chaA* as a control in our expression assays, but of the few pPRO::*chaA* transformants obtained, all contained amino acid mutations on the cytoplasmic face of the protein compared to the WT *chaA* sequence (S2 Fig). As pPRO24 is a multicopy plasmid, these data suggested that functional *chaA* is toxic to the cell at levels beyond WT *E. coli,* even on near-neutral (pH 7.2) media [30,59]. We did identify was the transporter, *zntB* (Table 1), which has previously been associated with $Zn^{2+}$ transport, but no studies have evaluated $Ca^{2+}$ binding in this transporter [46,47]. Overexpression of *zntB* showed a significant increase in carbonate precipitation compared to the overexpression of CA (Fig 5), but no significant difference was detected between induced and uninduced pPRO::*zntB,* even while an increase in expression was shown (S1 Fig). The $P_{prpB}$ promoter is leaky and as a trans-membrane protein, ZntB would have a longer half-life than cytoplasmic CA [61] that could allow ZntB to accumulate prior to induction (Fig 5). Alternatively, we may have exceeded the solute availability for $Ca^{2+}$ or $CO_2$ in the reaction, limiting maximal available $CaCO_3$ production.

The mutations in central metabolism appeared to affect carbonatogenesis through a combination of growth defects and electron transport (Table 1, Fig 1). The loss of Complex I (*nuoABCEFGHIJK*) dramatically reduced $CaCO_3$ precipitation across all substrates tested, even while the pH of the media was similar to WT (Fig 2). Complex I accepts electrons from $NADH_2$ to reduce ubiquinone [62,63]. The loss of Complex I does not prevent central metabolism from proceeding, as it can be compensated by a second NADH oxidoreductase complex (encoded by *ndh*). But this latter complex lacks the transmembrane component and does not translocate protons across the membrane [64,65], which could explain why Δ*nuoB* did not lower extracellular pH (Fig 2A) [66,67]. Nonetheless, carbonatogenesis was delayed in Δ*nuoB*, even when the pH exceeded the SI for $CaCO_3$ (Fig 2C), suggesting that the loss of Complex I may interfere with the onset of $CaCO_3$ precipitation, even when the required physiochemical conditions are met (Fig 2B and 2C).

We identified another key player in central metabolism that prevented precipitation: SQR (Table 1). A knockout in Δ*sdhC* dropped the pH considerably on all media examined (Fig 2A). Given that mutations in the *sdh* operon also interfere with electron transport, the reduced pH could be due to increased production of acetate [66,67]. Given its prominent role in cellular metabolism, the *sdh* operon is regulated by several factors, including the ferric uptake regulator (Fur) protein and we decided to explore the impact of Fe(III) amendment on carbonatogenesis, which accelerated $CaCO_3$ precipitation (Fig 4), possibly through the i upregulation of *sdh* expression, although these studies were carried out at room temperature rather than 37°C. Nonetheless, it is unlikely that the B4mSu media is Fe-limited and the identification of multiple genes involved in iron transport (Table 1) in our assays suggests that Fe impacts may extend beyond the expression of SQR. Additional continuous culture experiments using some of the Keio strains associated Fe metabolism would allow the exploration of this relationship further.

In addition to genes involved in electron transport, mutations were also identified in *lhgD* (*ygaF*) and *ubi* (Fig 1, Table 1). It is possible that limiting the entry of reducing equivalents to the quinone pool promotes overflow metabolism, which would otherwise lead to the production of acids, reducing pH [66,67]. Alternatively, the enzyme LhgD converts 2-hydroxyglutarate to α-ketoglutarate in the catabolism of lysine (Fig 1), which could affect amino acid metabolism and the release of $NH_4^+$ into the media. Given the need for amino acids in the precipitation media, $NH_4^+$ release may play a role in raising pH and promoting carbonatogenesis.

The MICP process using $Ca^{2+}$ homeostasis occurs with all calcium carboxylates tested, but is most pronounced with calcium acetate (in non-*E. coli* species) and calcium succinate [25,26,29]. Both acetate and succinate catabolism bypass the $CO_2$ liberated by the decarboxylation of α-ketoglutarate and succinyl-CoA, preventing $CO_2$ liberation in this step of the TCA cycle (Fig 6). In the case of acetate, this prevents regeneration of oxaloacetate, and the cell uses the glyoxylate

shunt to lyse isocitrate into succinate, with acetate added to glyoxylate via condensation to produce malate (Fig 6). Much of the $CO_2$ usually generated via the TCA cycle diffuses out of the cell, but a certain portion is used in anabolic pathways, such as lipid synthesis. Under such conditions, growth on calcium acetate is dependent on functioning CA to compensate for the lack of metabolic $CO_2$ produced [26]. A significant increase in MICP is also seen when *E. coli* expresses the CA genes *can* and *cynT* (Fig 5) on calcium succinate. Collectively, these results suggest our MICP model (Fig 6), which may be more dependent on $CO_2$ in the cellular environment than previously recognized [26].

Finally, it has been well established that cellular polymeric structures play a role in MICP, where precipitation appears to be driven by the initiation of nucleation of $CaCO_3$ crystals on the cell surface [68–70]. Such studies have shown that repeating polymers, such as peptidoglycan, teichoic acids, LPS, and EPS are able to coordinate bound $Ca^{2+}$, enabling $CaCO_3$ crystallization to proceed at lower chemical pressures and promoting precipitation [69–72]. Our work similarly suggests that such repeating structures play an important role in *E. coli,* but expands these polymers to include fimbriae and flagella (Table 1).

The conditions that promote MICP in *E. coli* ($CO_2$ uptake, $Ca^{2+}$ enrichment, increased pH, nucleation) actually reflects the conditions in caves that lead to secondary precipitation of $CaCO_3$ [73]; secondary deposits form in caves when surface water becomes saturated with $CO_2$ as it passes through the soil, creating a weak carbonic acid that dissolves the limestone rock ($CaCO_3$) and becomes saturated with $Ca^{2+}$ and $CO_3^{2-}$ [73]. In the atmosphere of the cave, the off-gassing of this $CO_2$ raises the pH and dramatically increases the SI for $Ca^{2+}$ and $CO_3^{2-}$, leading to precipitation [73]. Our bacterial system is the spatial inverse of this precipitation (Fig 6): the $CO_2$ is taken up by the cell via CA to compensate for the reduced cytoplasmic $CO_2$, while $Ca^{2+}$ levels rise through cellular export while amino acid metabolism increases the pH through the production of $NH_4^+$. The sum of these processes leads to a dramatic increase in the SI for $CaCO_3$ in the microenvironment surrounding the cell (Fig 6). These improvements to our existing MICP model underscore the intricate relationship intricate relationship between microbial metabolism, structure, and mineral nucleation [26]. It also identifies a myriad of targets in *E. coli* that could be altered to increase the $CaCO_3$ yield, including: increasing the cellular uptake of $CO_2$; increasing the export of $Ca^{2+}$; increasing the generation of $NH_4^+$ through amino acid metabolism; and enriching the presence of MICP-promoting polymers around the cell. Nonetheless, the results do raise additional questions: what is the role of Fe(III) in promoting MICP; does ZntB serve as a Ca2+ transporter in *E. coli;* and the potential for other polymeric structures, such as fimbrae and pili to serve as nucleation sites. Further experiments will endeavor to answer these questions and further refine our model, with a goal of increasing the potential for this method to be utilized in the production of industrially relevant carbonates without the need for ureolysis.

## Supporting information

**S1 Table. Primers used in this study and supplemental figure S1.** These primers were used to amplify genes for cloning into the pPRO24 plasmid. Other listed primers were used to sequence those constructs.
(XLSX)

**S1 Fig. SDS-PAGE gel of *E. coli* overexpression strains.** Slide 1 shows soluble fractions of induced and uninduced expression strains. All cultures were grown in 50 LB media supplemented with 100 µg/mL ampicillin at 37°C while shaking. A "+" indicates the culture was induced with 2.5 g/L sodium-propionate when the $OD_{600} = 0.6$. Following induction, flasks were moved to a room temperature shaker, allowed to grow overnight, and then harvested the next morning. A "-" indicates the flask was not induced. DE3 contains the empty pPRO24 vector, SKP004 contains the *can* expression plasmid, and SKP018 contains the *zntB* expression plasmid. The total amount of protein loaded into each lane is listed at the top. The box highlights overexpression of Can in the SKP004 strain. The first lane contains 10 µL PageRuler ladder, with the band sizes labeled on the left. Slide 2 shows insoluble fractions of induced and uninduced expression strains.
(TIF)

**S2 Fig. Protein alignment of ChaA.** Only three ChaA transformants were obtained from the the pRPO::*cha*A expression construct transformed into *E. coli*. Prior sequence analysis has suggested that E85 (green) is involved in proton exchange [1] and the central acidic region between E199 – D208 (blue) is a $Ca^{2+}$ binding domain [2]. While Ivey et al. (1993), suggested the protein contained 11 transmembrane a-helices [2], all but one (L17 – P37) overlap with the 10 domains identified by a Phyre2 structural analysis (grey) [3]. Sequencing of the *chaA* gene inserts revealed the following mutations: L221V, A236E, and S244P, all of which were associated with the sixth transmembrane domain.
(PDF)

## Author contributions

**Conceptualization:** Kathleen R. Gisser, Hazel A. Barton.

**Data curation:** Matthew Edward Jennings, Hazel A. Barton.

**Formal analysis:** Matthew Edward Jennings, Reilly S. Blackwell, Hazel A. Barton.

**Funding acquisition:** Kathleen R. Gisser, Hazel A. Barton.

**Investigation:** Matthew Edward Jennings, George J. Breley, Reilly S. Blackwell, Anna Drabik, Joe Kainrad, Victoria Ligman, Jaycie Proctor, Rose Deshler, Brian A. O'Hart, Arlyn Rivera, Lorelei Centrella, Sara Bonn, Bisma Chaudhry, Hazel A. Barton.

**Methodology:** Matthew Edward Jennings, George J. Breley, Reilly S. Blackwell, Anna Drabik, Hazel A. Barton.

**Project administration:** Matthew Edward Jennings, Kathleen R. Gisser, Hazel A. Barton.

**Resources:** Kathleen R. Gisser, Hazel A. Barton.

**Supervision:** Hazel A. Barton.

**Validation:** Matthew Edward Jennings.

**Visualization:** George J. Breley, Reilly S. Blackwell, Hazel A. Barton.

**Writing – original draft:** Matthew Edward Jennings.

**Writing – review & editing:** Matthew Edward Jennings, George J. Breley, Reilly S. Blackwell, Anna Drabik, Kathleen R. Gisser, Victoria Ligman, Jaycie Proctor, Rose Deshler, Arlyn Rivera, Sara Bonn, Bisma Chaudhry, Hazel A. Barton.

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
