## [Decision Letter · Decision Letter 0]

24 Apr 2025

Thank you for submitting your manuscript to PLOS ONE. After careful consideration, we feel that it has merit but does not fully meet PLOS ONE’s publication criteria as it currently stands. Therefore, we invite you to submit a revised version of the manuscript that addresses the points raised during the review process.

We look forward to receiving your revised manuscript.

Kind regards,

Sreenivasulu Basha, Ph.D

Academic Editor

PLOS ONE

[This work was supported by the Defense Advanced Research Projects Agency: Engineered Living Materials grant HR0011-18-9-0007 and NSF grant IIP-2122799.]

[HB, KG Grant# HR0011-18-9-0007

DARPA ELM: Engineered Living Materials

https://www.darpa.mil/research/programs/engineered-living-materials

The funders had no role in the study design, data collection and analysis, decision to publish, or preparation of the manuscript.

HB, KG, MJGrant# IIP-2122799

National Science Foundation PFI Grant

https://new.nsf.gov/funding/initiatives/pfi

The funders had no role in the study design, data collection and analysis, decision to publish, or preparation of the manuscript.]

Reviewers' comments:

Reviewer's Responses to Questions

**Comments to the Author**

1. Is the manuscript technically sound, and do the data support the conclusions?

Reviewer #1: Partly

Reviewer #2: Yes

2. Has the statistical analysis been performed appropriately and rigorously?

Reviewer #1: N/A

Reviewer #2: Yes

3. Have the authors made all data underlying the findings in their manuscript fully available?

Reviewer #1: Yes

Reviewer #2: No

4. Is the manuscript presented in an intelligible fashion and written in standard English?

Reviewer #1: Yes

Reviewer #2: Yes

Reviewer #1: I have reviewed the manuscript titled, “The diversity of cellular systems involved in carbonate precipitation by Escherichia coli”. The authors clearly lay out the importance of finding new ways to trap CO2 and present one possible route – MICP. They identified genes possibly involved in the E. coli MICP pathway, classified into four primary groups, and further investigated some of the identified genes to better understand their possible role. While the bulk of the manuscript has sound conclusions, I still have some lingering questions, especially about the iron metabolism and efflux pump results and conclusions, therefore I would recommend this manuscript for minor revisions. Below are my points of concern.

• Some of the tables and figures need some further clarification in their captions. The Table 1 caption should include that the genes that are highlighted in gray and bolded are those that underwent further investigation. Additionally, it would be useful if Table 1 had another column which stated if the carbonate production was higher or lower in that strain. Figure 2C should indicate that the dash line is the pH at which SI is reached. Figure 4 should clarify what the different colors are in the two graphs and if only the WT strain was used in these studies.

• In the section on central metabolism, how were the representative genes selected for further study over the others? It would be useful to have a brief explanation for this. The authors look at plating versus liquid culturing methods, but what calcium source is used in the liquid cultures shown in Fig. 2?

• When focusing in on complex I, the authors conclude that the data in Fig. 3 and Fig. 2 are correlated, but this is not a true statement - there are conflicting results between Fig. 2A and Fig. 3. For example, in Fig. 3 for both acetate and propionate, the level of insoluble Ca2+ is very low and almost indistinguishable for all three strains tested. However, according to Fig. 2A, levels of the nuoB knockout were noticeably less than the other two strains. Please check the color-coding scale in Fig. 2A to ensure that the results between liquid and solid culturing do actually correlate.

• The iron metabolism section is underdeveloped in my opinion. The authors nicely tested the shift in timing of CaCO3 precipitation with the WT strain, but did not test any of the strains with the knockouts that they have found may be involved. Since three of the genes found are transporters, this would have helped to solidify a link between iron transport and CaCO3 precipitation. The authors should either perform this experiment or comment on why this was not tested.

• Based on what is shown in the efflux pumps section, I am not convinced that zntB is the likely Ca2+ exporter involved in the MICP pathway. The data shown in Fig. 6 for the control and yrbG/can/cynT does nicely show the validity of their expression approach, however it does not conclusively show that zntB is functioning as a Ca2+ transporter. The levels should not have already been that high for uninduced zntB. The explanation about the promoter being leaky is not a proper explanation for this observation considering the other plasmids incorporated the same promoter and this issue was not encountered. Further explanation needs to be included here. Additionally, zntB cannot be named as THE exporter when the other two efflux pumps identified weren’t even tested in the same fashion just because less was known about these two pumps. Please comment on the lack of testing of the other two pumps.

Reviewer #2: Calcium stress-dependent Microbially Induced Carbonate Precipitation (MICP) in E. coli is a novel and promising approach for trapping atmospheric CO2 as calcium carbonate (CaCO₃). Unlike ureolysis-based MICP, this system avoids ammonia-related issues, offering a cleaner alternative for industrial applications; however, the yield of precipitated CaCO3 is presently quite low.

With the aim to optimize the process, the authors of the present study screened a E. coli knockout library and identified 54 genes whose inactivation affects CaCO3 precipitation. These genes are involved in central metabolism, iron metabolism, cell architecture, and ion transport systems.

General comments

The manuscript is generally well written and detailed, although several paragraphs in the Results and Discussion sections are a little “verbose”: overall, the MS may benefit from a more concise style.

Titles of the Results sub-sections are appropriate, but quite generic and scarcely informative.

The work is well done from a technical perspective: experiments are well conceived and generally well performed (with few exceptions: see below). Data reported generally support the Authors’ conclusions and speculations. Still, most issues are not examined in great detail.

The graphic quality of the figures is generally adequate, with complete and descriptive legends (with minor exceptions: see below).

In the text there are way too many references to data not shown, which in my opinion should be included as Supplementary Materials.

Methods are well detailed.

The relevant Literature is cited in the appropriate context.

The novelty degree of the paper is not overwhelming, but is nonetheless significant, since it identifies several new potential genetical targets/cellular processes which may be manipulated to enhance CO2 fixation by MICP in E. coli: namely, i) CO2 uptake; ii) Ca2+ efflux; iii) ammonium generation by amino acids metabolism; iv) enrichment of MICP-promoting polymers around the cell surface.

Minor Remarks

• The KO E. coli library was screened by a rapid qualitative plate visual assay to evaluate any genetic impacts on carbonatogenesis, allowing the identification of 54 mutants with altered CaCo3 precipitation phenotype. However, only the list of these mutants is reported in Table 1, with no results clarifying to which extent carbonatogenesis was deficient (or possibly enhanced) in these strains.

• Several representative mutants (indicated in bold in Table 1) were selected for further characterizations, including (for some but not all the strains) a more quantitative assay in liquid medium to precisely evaluate the amount of precipitated carbonate (Fig. 3 and Fig. 5). It would be useful if the quantitative assays were performed for all the selected mutants.

• In fig 2B-C, the authors claim that the onset of carbonate precipitation in liquid medium occurs when a notable bump in OD600 and a simultaneous drop in the pH of the culture are registered. However, these bumps/drops are quite modest for most strains (including the wild type, dark blue symbols), at least in this particular experiment, whereas they are much more evident in fig. 4.

Role of Iron metabolism.

• The authors list several mutants in iron metabolism with reduced CaCO3 precipitation, but they report no qualitative or quantitative analysis about the amount of this reduction. These mutant were no further characterized.

• The role of iron on the timings of carbonatogenesis in liquid medium was evaluated by continuously measuring the pH medium (with the onset of CaCO3 crystal accumulation coinciding with a sudden drop in pH). The experiment was performed at 24°C (lower temperatures reduce growth rate and improve temporal resolution of the assay). In other experiments (e.g. fig. 2) cell are grown at 37°C. Has temperature any impact at all on the efficiency of CaCO3 precipitation?

• It is unclear what the series plotted in the upper (untreated) and lower (+FeCl3) panels of Fig4 represents: various experimental replicates for wild type strains? Different strains? Nothing is specified, either in the legend or in the main text.

• Addition of FeCl3 to growth medium clearly anticipates the timings of carbonatogenesis. Does it also affect the amount of precipitated CaCO3?

• Can FeCl3 supplement improve carbonatogenesis in mutants deficient in iron metabolism?

Efflux pump

• Consistently with the key role of calcium export in carbonatogenesis, the authors identified in their preliminary screen several ion efflux pumps which may drive the process, in particular Zntb, which was previously involved in zinc homeostasis (whereas no Ca2+ binding has been reported so far in literature). Unfortunately, no quantitative data for carbonatogenesis relative to the ΔzntB null mutant are shown in the manuscript.

• Instead, the authors convincingly show how the overexpression of zntB from a multicopy plasmid under an inducible promoter significantly increased (3-fold) CaCo3 precipitation, whereas no effect is observed upon overexpression of yrbG (a known calcium transporter not involved in carbonatogenesis: Fig. 5). However, no differences were registered for zntB between the induced and uninduced conditions, which may result from the leaky promoter coupled with a multicopy plasmid, as suggested by the author. Nonetheless, the dosing effect of induction on CaCO3 precipitation is clearly detectable when carbonic anhydrases are overexpressed using the same combination of plasmid and promoter: in this case, the difference may be due to the longer half-life of ZntB (a membrane protein) relative to the shorter half-life of the cytoplasmic carbonic anhydrases. Since SDS-PAGE expression data are cited but not shown, it is unclear which levels of overexpression are actually achieved for Zntb under uninduced/induced conditions.

• Did the authors test a low copy plasmid or another inducible promoter for their expression system?

**Do you want your identity to be public for this peer review?** For information about this choice, including consent withdrawal, please see our Privacy Policy

Reviewer #1: No

Reviewer #2: No

---

## [Author Response · Author response to Decision Letter 1]

9 Jul 2025

Please see attached Response to Reviewers file.

---

## [Decision Letter · Decision Letter 1]

2 Dec 2025

The diversity of cellular systems involved in carbonate precipitation by Escherichia coli

PONE-D-25-05958R1

Dear Dr. Matthew Edward Jennings,

We’re pleased to inform you that your manuscript has been judged scientifically suitable for publication and will be formally accepted for publication once it meets all outstanding technical requirements.

Kind regards,

Sreenivasulu Basha, Ph.D

Academic Editor

PLOS ONE

Additional Editor Comments (optional):

Dear Authors,

Thank you for your thorough and timely responses to the reviewers’ comments. We are pleased to inform you that your manuscript has been accepted. Congratulations on your excellent work.

Reviewers' comments:

Reviewer's Responses to Questions

**Comments to the Author**

Reviewer #1: All comments have been addressed

Reviewer #3: All comments have been addressed

2. Is the manuscript technically sound, and do the data support the conclusions?

Reviewer #1: Yes

Reviewer #3: Yes

3. Has the statistical analysis been performed appropriately and rigorously?

Reviewer #1: Yes

Reviewer #3: N/A

4. Have the authors made all data underlying the findings in their manuscript fully available?

Reviewer #1: Yes

Reviewer #3: Yes

5. Is the manuscript presented in an intelligible fashion and written in standard English?

Reviewer #1: Yes

Reviewer #3: Yes

Reviewer #1: (No Response)

Reviewer #3: (No Response)

**Do you want your identity to be public for this peer review?** For information about this choice, including consent withdrawal, please see our Privacy Policy

Reviewer #1: No

Reviewer #3: No

---

## [Editor Report · Acceptance letter]

PONE-D-25-05958R1

PLOS One

Dear Dr. Jennings,

I'm pleased to inform you that your manuscript has been deemed suitable for publication in PLOS One. Congratulations! Your manuscript is now being handed over to our production team.

Kind regards,

on behalf of

Dr. Sreenivasulu Basha

Academic Editor

PLOS One